# *TERT* Promoter Mutations are Associated with Visceral Spreading in Melanoma of the Trunk

**DOI:** 10.3390/cancers11040452

**Published:** 2019-03-30

**Authors:** Simona Osella-Abate, Luca Bertero, Rebecca Senetta, Sara Mariani, Francesco Lisa, Vittoria Coppola, Jasna Metovic, Barbara Pasini, Susana Puig S, Maria Teresa Fierro, Esperanza Manrique-Silva, Rajiv Kumar, Eduardo Nagore, Paola Cassoni, Simone Ribero

**Affiliations:** 1Department of Medical Sciences, Pathology Unit, University of Torino, 10126 Torino, Italy; simona.osellaabate@unito.it (S.O.-A.); luca.bertero@unito.it (L.B.); sara.mariani@unito.it (S.M.); vittoria.coppola8@gmail.com (V.C.); jasna.metovic@unito.it (J.M.); 2Pathology Division, “Città della Salute e della Scienza di Torino” University Hospital, 10126 Torino, Italy; rsenetta@cittadellasalute.to.it; 3Department of Medical Sciences, Dermatology Unit, University of Torino, 10126 Torino, Italy; francesco.lisa@unito.it (F.L.); mariateresa.fierro@unito.it (M.T.F.); simone.ribero@unito.it (S.R.); 4Department of Medical Sciences, Medical Genetics Unit, University of Torino, 10126 Torino, Italy; barbara.pasini@unito.it; 5Melanoma Unit, Dermatology Department, Hospital Clinic, Universitat de Barcelona & Institut d’investigacions biomèdiques August Pi i Sunyer (IDIBAPS), 08036 Barcelona, Spain; susipuig@gmail.com; 6Servicio de Dermatologia, Instituto Valenciano de Oncología, 46009 Valencia, Spain; emanriques19@gmail.com (E.M.-S.); eduardo_nagore@ono.com (E.N.); 7Division of Molecular Genetic Epidemiology, German Cancer Research Center, 69120 Heidelberg, Germany; r.kumar@dkfz.de

**Keywords:** *TERT* promoter, trunk, melanoma, visceral metastases

## Abstract

Survival predictions are currently determined on the basis of *NRAS/BRAF* mutations, even though *TERT* promoter mutations have been recently associated with a poor prognosis in stage I-II melanomas. Usually, it is not recommended to perform a mutational test on primary melanoma, as the results do not always reflect the mutational status of metastases. In particular, trunk melanomas have been reported to have an unfavourable prognosis. A series of 105 advanced melanoma patients were analysed by *TERT* promoter Sanger sequencing. Univariate/multivariate binary logistic regression models were performed using progression to a visceral site as the dependent variable and patient/tumour characteristics as covariates. Performance of the model was assessed in an external independent primary melanoma patients’ dataset. Male gender (odds ratio (OR), 344; 95% CI, 1.12–10.6; *p* = 0.031), AJCC (American Joint Committee on Cancer) classification (OR, 022; 95% CI, 0.07–0.67; *p* = 0.008), SLNB (Sentinel Lymph Node Biopsy) status (OR, 3.05; 95% CI, 1.06–8.78; *p* = 0.039) and *TERT*-mutated trunk lesions (OR, 3.78; 95% CI, 1.35–10.6; *p* =  0.011) were significantly associated with the risk of developing a visceral spreading as first site of progression using multivariate logistic regression analysis. These results were confirmed in the external validation control group. Therefore, in trunk primary melanomas, due to their high risk of progression to visceral sites, we encourage somatic *TERT* mutation analysis at diagnosis to identify those patients who would potentially benefit from a more intensive follow-up protocol and a prompt initiation of therapy.

## 1. Introduction

Genetic testing for targetable somatic mutations is considered mandatory by the European Guidelines in the context of diagnosis, treatment and follow-up of cutaneous melanomas in patients with advanced disease (unresectable stage III or stage IV), and highly recommended in high-risk resected disease (stage IIC or stages IIIB–IIIC) [1]. In *BRAF* wild type tumours, alternative mutations occurring at *NRAS* and *c-Kit* genes must be tested. Tumour heterogeneity in advanced stage melanoma has important implications for molecular testing and treatment. Mutation load increases with progression and unique patterns of genetic changes termed ‘evolutionary trajectories’, have been observed in different melanoma subtypes [2,3]. In particular, as reported by Shain et al., melanomas mutated at *NRAS* or *BRAF* (V600K or K601E) are more commonly associated with intermediate lesions or melanomas in situ that have already accumulated other pathogenic mutations [2]. In clinical practice, it is not recommended to test for *BRAF* mutations in primary cutaneous melanoma unless these data are required to guide systemic therapy in advanced stages. Several studies suggest that detection of a wild type *BRAF* (B-Raf proto-oncogene) in the primary tumour may not necessarily reflect the *BRAF* mutation status of metastases [4,5]. Therefore, clinicians should consider repeating the test in the most recent metastasis (if available) to determine the *BRAF* mutation status [4]. In daily clinical practice, patients often come to our attention with advanced stages, progressing either after a positive or a negative sentinel lymph node (SLN), or directly from a primary to a distant metastatic site [6,7]. Moreover, when metastasis sampling is not feasible due to the anatomical site or to other patient clinical comorbidities, mutation analysis is performed on the primary lesion.

Recently, Nagore et al. provided preliminary evidence that in stage I/II melanoma patients *TERT* promoter status in combination with *BRAF*/*NRAS* mutations can be used to identify patients at risk of aggressive disease and the possibility of further prognostic refinement by assessing the rs2853669 polymorphism within *TERT* promoter [8].

Based on these assumptions, we analysed a consecutive series of patients with advanced melanoma who underwent mutational testing preliminary to treatment with specific targeted therapies at the tertiary Dermo-Oncologic Centre of our Hospital. In particular, in this study we retrospectively analysed advanced melanomas that had directly progressed from stage IB/II AJCC (American Joint Committee on Cancer) 2017 [9], whose mutational status of *BRAF*, *KRAS*, *NRAS* and *PIK3CA* genes had already been tested. In these specimens, *TERT* promoter status and its rs2853669 single nucleotide variant (SNV) were then investigated to ascertain associations between the molecular profile and the pattern of progression with the aim of determining the prognostic value of *TERT* promoter mutations in a subset of IB/II stage melanoma patients with poorer prognosis.

## 2. Results

### 2.1. Clinical Characteristics

A total of 105 patients (Table 1) satisfied the inclusion criteria of our study protocol. The average Breslow thickness of primary melanomas was 3.7 mm ± 2.8, while the median follow-up after melanoma diagnosis was 5.8 years (3.2–12.2). The median age at diagnosis was 70 years (range 22–88). Overall, 63 patients (60.0%) were male. The majority of melanomas appeared on the trunk (*n* = 41, 39.1%). The Nodular melanoma (NM) histotype was observed in 16 out of 105 (15.2%). Ulceration was present in 49 out of 105 patients (46.7%). SLNB (Sentinel Lymph Node Biopsy) was performed in 41 out of 105 patients (39.0%). In the remaining 64 patients SLNB was not performed: in 27 patients after collegial decision, in six patients whose melanoma thickness was above 4 mm and between 0.8–1 mm respectively (SLNB not indicated, ante 2018), in 16 patients older than 75 years, in seven patients with other comorbidities, in two patients whose SLN were not traced, in four patients refusing the procedure and in two patients diagnosed ante-SLNB technique.

Metastases included in the study appeared after median follow-up of 1.9 years from primary melanoma diagnosis (0.7–3.8). Fifty-eight patients developed regional metastases only (24 skin, 29 lymph nodes, and five skin and lymph nodes), 15 had concomitant regional and distant metastases and 32 exhibited visceral metastases as first and unique sites of relapse. In detail, among the latter with visceral metastases, 19 patients had a single site of involvement (eight lung, seven brain, two liver, one spleen and one peritoneum), nine patients exhibited two metastatic sites and four patients had more than two sites involved. The visceral pattern of relapse seems to be related to the trunk site of primary melanoma but not with NM histotype. No differences in median time to relapse were observed to be associated with the site of disease relapse (log-rank test *p* = 0.9564) (Table 1, Figure 1).

### 2.2. Mutational Assessment

Mutational assessment to guide targeted therapy was performed in 42 primary lesions: 20 patients (48.8%) with visceral metastases were surgically inaccessible, 8 had regional lymph node and visceral metastases without indications for lymph node dissection and 14 had only regional metastases that did not undergo surgical resection due to other clinical conditions (older age, comorbidities). In the remaining patients, mutational status was assessed on the most recently available metastases: 46 regional metastases and 17 visceral metastases, respectively (Table 1). All patients were tested with Sequenom (32 WT (Wild-Type), 44 *BRAF*-mutated, 27 *NRAS*-mutated, 1 *KRAS*-mutated and 1 *PIK3CA*-mutated), among these 4 out of 6 acral melanomas (4 WT, 1 *NRAS*-mutated and 1 *BRAF*-mutated) were also tested for *KIT* mutations: only 1 out of 4 WT lesions carried a D816H *KIT* mutation. Moreover, 74 out of 105 samples showed the most frequent *TERT* promoter mutations: −124 C > T or −146 C > T (minor mutations were reported in Table 2).

No association between *BRAF*, *KRAS*, *NRAS*, *PIK3CA* and *TERT* promoter mutations or rs2853669 was observed with first relapse site. Conversely, the *TERT* promoter mutation detected in any tested site was associated with the trunk site of primary lesion (34 out 41 patients, 82.9%) (Table 1 and Table 3).

In consideration of the limited sample size due to a real-life study setting, the two most represented mutations (−124 C > T and −146 C > T) have been combined together for the analyses. Indeed, to analyse the variants separately it would have been necessary a minimum of 230 melanoma cases in order to detect a significant 5% difference in outcomes between −124 and −146 *TERT* mutations according to a power calculation analysis using visceral specific recurrence as the primary end point with a 80% power and a two-sided α = 0.05 test. Moreover, we have not stratified patients according to −57A > C because only two cases were observed (Table 2).

### 2.3. Association Between Trunk Site and TERT Mutation

Considered that visceral spread seemed to be associated with a trunk primary melanoma and that the *TERT* promoter mutation frequently occurred in patients whose primary lesion developed at this site, we analysed the progression pattern by evaluating the association between the trunk site and the *TERT* promoter mutation. In the group in which the mutational status was performed on the primary lesion, data showed that 12 out of 18 cases (66.7%) with a −124 or −146 C > T mutation and the primary site on the trunk had developed visceral spreading as the first site of progression.

At univariate analysis (Table 4), logistic regression highlighted a significant association between the *TERT* promoter mutation and trunk site with visceral spreading (OR 5.33, CI 1.02–27.7) when considering cases in which only the primary lesion was analysed. This finding was consistent even when considering the cases in which both the primary lesion and the metastasis were analysed. In summary, 19 out of 34 patients (55.9%) who had a *TERT* promoter mutation in the primary or metastatic lesion associated with a primary melanoma trunk site, developed visceral spreading as the first site of progression (OR 3.64, CI 1.14–11.66). The other variables associated with visceral spreading were male gender, trunk site, performed SLNB procedure (thus correctly staged) and also lower AJCC stage. 

Multivariate logistic regression analyses (Table 5) confirmed the significant association between *TERT* promoter mutation and trunk site with visceral spreading also when adjusting for age, gender, AJCC staging and SLNB, suggesting that preferential visceral spreading as the first site of progression is related to *TERT*-mutated lesions in patients whose primary lesion is located on the trunk. Moreover, the Hosmer–Lemeshow goodness-of-fit statistics (*p*-value = 0.5460) indicated that the model adequately describes the data.

### 2.4. Validation in the Independent Cohort of the Instituto Valenciano de Oncologia Data Set

To test the reproducibility of the association between *TERT* promoter mutation in trunk site with visceral metastases as first site of progression, we investigated its performance in data collected from 83 stage II primary melanoma patients all progressed to a metastatic stage recruited in Valencia by the Instituto Valenciano de Oncologia (Prof. E Nagore) (Table 6). Data on the variables used to define association were extracted from dataset and equally classified as for the pooled data described above. Multivariate logistic regression analyses performed in the Valencia dataset (Table 7) confirmed the significant association between *TERT* promoter mutation and trunk site with visceral spreading (OR 4.81, CI 1.01–22.9; Hosmer–Lemeshow goodness-of-fit statistics, *p* = 0.8986).

## 3. Discussion

Our experience in daily practice highlights that mutational status assessment of progressed Stage IB/II melanoma patients is conditioned not only by the progression site, but also by patient comorbidities that can influence surgical management. All these conditions affect the available tissue type for molecular testing. In particular, disease progression in not surgically accessible anatomical sites often (40% of cases) makes it necessary to use the primary lesion instead of the most recently developed distant metastasis. Guidelines for follow-up of stage I–II patients (correctly staged through SLNB or not) differ around centres and usually do not envisage serial screening of visceral sites in the low risk of progression group [10,11] although many patients with an intermediate Breslow thickness melanoma risk progress directly to a visceral site [6,12]. In stage IB/II patients, a more individualised follow-up taking into account the risk factors associated with relapse in visceral sites, which include male gender, trunk site and different SLNB proposal and management [6,7,13,14,15,16], would be desirable.

In our consecutive case series of metastatic melanomas, stage IB was more commonly associated with visceral site than higher stages (IIA, IIB and IIC). At the moment, no reliable biomarkers except SLN status have proven to be associated with the real risk of primary melanoma progression, and no information can be specifically ascertained based upon the hypothesis (not supported by biological data, so far) of a predisposition of trunk melanoma to progress to visceral sites [14,17].

In this context, new markers are needed for discriminating patients at higher risk of visceral relapse. Despite the heterogeneity of our cohort, our results showed that the *TERT* promoter mutations in primary trunk melanoma are related to visceral spreading as the first site of progression, independently of SLNB management and AJCC stage, thus suggesting that −124 or −146 C > T mutations may be considered as risk factors of visceral spreading in trunk lesions. This finding has been validated in an independent Valencia cohort of primary melanoma patients all progressed to a metastatic phase, confirming the reliability of this association.

Furthermore, the association between *TERT* promoter mutations and trunk site is supported by evidence that melanomas occurring in intermittently sun-exposed skin as trunk site displayed an increased prevalence of *TERT* promoter mutations compared with melanomas occurring in sun-protected areas [18].

It is well known that *TERT* mutations enhance the expression of the *TERT* gene by creating *de novo* binding motifs for different transcription factors [19] involved in tumorigenic mechanisms, but its prognostic role is debated. Several authors have reported a prognostic significance of *TERT* promoter mutations in primary melanomas [18,20,21], whereas other studies [22,23] found no impact in primary and metastatic melanoma. In our experience, *TERT* promoter mutations have been identified both in primary and metastatic visceral lesions of patients with a trunk primary. These data are in agreement with Shain et al. [2], who also identified *TERT* promoter mutations in intermediate lesions and melanomas *in situ*, in addition to invasive and metastatic melanomas, thus confirming that *TERT* mutations are selected in an early stage of the neoplastic progression and maintained during the entire metastatic process [2], in variance with other mutations like *BRAF* which can show a discordance between primary and metastatic tumours in 11% of cases [4,24].

A meta-analysis indicates that *TERT* promoter mutations are associated with patient age, gender and distant metastasis in individuals with cancers [25]. In particular promoter mutations were found to be independent risk factors for distant metastases in thyroid carcinoma. Moreover, in thyroid carcinoma *TERT* inhibition has been related to reduced cell growth, invasion, migration and angiogenesis [26,27]. Trunk melanomas have been associated in many studies to visceral involvement [15,16,17], *TERT* promoter mutation can be the biological explanation of trunk melanoma ability to skip regional metastases by promoting visceral spreading.

Our data reflect daily clinical practice, which often deviates from optimal conditions. We are aware of the limitations of this study:

(1) The cohort is not homogeneous in term of SLN management, but in order to keep this fact in consideration, the model had been adjusted for this feature (SLNB performed vs. not performed). The association between visceral progression and *TERT* promoter mutation in trunk site patients should be considered even more reliable due to the fact that it maintained its prognostic role after this adjustment. 

(2) The relative low sample of our study did not allow us to perform stratification on the basis of *TERT* promoter mutation variant.

(3) It was not possible to confirm the preservation of *TERT* promoter mutation along the disease course (from primary tumour to metastases in the same patient) due to incomplete specimen availability.

(4) Conventional Polymerase Chain Reaction (PCR) followed by Sanger sequencing can underestimate the wild type/mutated allele ratio of *TERT* promoter compared to other methods [28].

However, our results and the external validation in a group of progressed primary melanomas support the previously reported finding of a worse outcome in *TERT*-mutated trunk melanomas. Therefore, we encourage *TERT* promoter mutation analysis at diagnosis in the primary melanomas of the trunk since their progression will likely occur in a visceral site in the majority of cases and patients would potentially benefit from early detection and prompt treatment. 

## 4. Material and Methods

### 4.1. Patients

A series of 105 progressed stage IB/II AJCC melanoma patients (24 stage IB, 31 stage IIA, 29 stage IIB and 21 stage IIC) with complete clinicopathological annotated features underwent mutational analyses for therapeutic purposes between April 2014 and December 2017 at the Pathology Unit and were followed up at the Dermatologic Clinic of “Città della Salute e della Scienza” University Hospital (Torino, Italy).

The study was conducted in accordance with The Code of Ethics of the World Medical Association (Declaration of Helsinki) for experiments involving humans and within the guidelines and regulations defined by the Research Ethics Committee of the University of Turin. This study was approved by the Research Ethics Committee of the University of Turin (DSM-ChBU 5/2016). Considered the retrospective nature of the research protocol and the lack of clinical impact on patient care, no specific written informed consent had been required.

Clinical, epidemiological and histological data were collected from the medical history of patients, whose tumours were diagnosed, treated and followed up according to previously reported protocols [6,7,25].

At our institution, SLN biopsy (SLNB) has been performed since 1998 and the criteria adopted for SLNB inclusion have been previously reported [13,14,15]. Age above 75 years and significant comorbidities are exclusion criteria for this procedure. Moreover, due to the lack of specific guidelines, in thick (>4 mm) melanomas [15,16], a multidisciplinary team discusses each case analysing the risk/benefit ratio before proceeding with SLNB. All decisions are based upon the physicians’ experience and clinical situation, considering that no evidence-based recommendations are available in this setting. All patients gave their written consent before undergoing ultrasound or CT scan to exclude the presence of regional of distant metastases before SLNB or at diagnosis according to regional guidelines (http://www.reteoncologica.it).

### 4.2. Mutational Status and TERT Promoter Assessment 

Primary or metastatic tumour tissue sections were obtained for DNA extraction as previously described [29]. Mutational detection was performed using the Sequenom MassARRAY^®^ system (Sequenom, San Diego, CA, USA) in conjunction with the Myriapod Colon Status kit that identifies 58, 54, 23 and 66 nucleotide substitutions in the *KRAS*, *NRAS*, *BRAF* and *PIK3CA* genes, respectively. Mutant and wild type alleles were discriminated with the Sequenom MassARRAY^®^ Analyser 4 platform. When warranted, the *KIT* mutational status was assessed by Sanger sequencing in *BRAF* wild type samples (primary acral or mucosal lesions) [30]. Mutational status of the *TERT* core promoter was determined in samples by PCR and Sanger sequencing between nucleotides −27 and −286 from the starting region (ATG) of the coding site, which includes the polymorphic site represented by rs2853669.

In detail, we amplified the *TERT* promoter (located on chromosome 5) target region with a touch-down PCR scheme using TaqGold 360 + GC enhancer (Applied Biosystems, Waltham, MA, USA) and the following primer pair; *TERT* promoter forward 5′-CTCCCAGTGGATTCGCGGGC-3′ and reverse 5′-CCCACGTGCGCAGCAGGAC-3′, as described by Heidenreich et al. [31]. With 2% agarose gel electrophoresis, we verified that the PCR products were of the expected size (260 bp) and free of aspecific amplicons. PCR products were then purified and used as templates for the sequencing reactions, which were performed in both directions (forward and reverse) with the same primers using the BigDye Terminator v1.1 Cycle Sequencing Kit (Applied Biosystems). After purification, the sequences were analysed by Sanger direct sequencing using the ABI PRISM 3130 Genetic Analyzer (Applied Biosystems), with the Sequencing Analysis 5.0 Software (Applied Biosystems). The two most frequently identified variations within the *TERT* promoter gene region at positions 1295228 and 1295250 are known as C228T and C250T, respectively. These mutations are located at −124 and −146 bp upstream of the ATG start codon and were considered for analyses.

### 4.3. Statistical Analyses

All analyses were performed using Stata/MP 15.0 Statistical Software (STATA, College Station, TX, USA). Continuous variables were summarised as the mean and standard deviation (SD), whereas for categorical variables the frequency was provided. The patients’ characteristics were compared using the chi-squared test for categorical variables and the T-test or ANOVA test for continuous variables, according to Bonferroni corrections. Univariate/multivariate binary logistic regression models were performed using progression to a visceral site as the dependent variable (yes or no) and patient/tumour characteristics as covariates. Odds ratios and 95% CIs were estimated. The Hosmer–Lemeshow goodness-of-fit statistics were used to determine whether the model adequately described the data.

Disease-free interval (DFI) was calculated from the date of primary lesion diagnosis to the date of tumour progression/recurrence or last follow-up. Survival curves between different groups were plotted using the Kaplan–Meyer method and the statistical comparisons were performed with log-rank test.

The final association (*TERT* promoter mutation and Trunk site towards visceral progression) has been validated in an external independent cohort of primary melanoma patients.

## 5. Conclusions

Our findings encourage *TERT* mutation analysis at diagnosis in primary melanomas arising on the trunk since they are more likely progress to visceral site. *TERT* screening would help in selecting patients who will potentially benefit from a more intensive follow-up protocol and a prompt initiation of therapy.

## Figures and Tables

**Figure 1 cancers-11-00452-f001:**
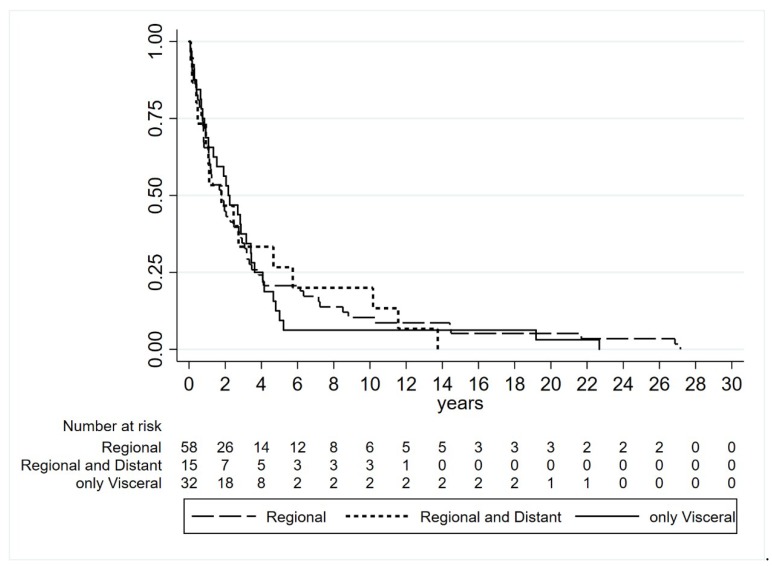
Time to relapse based on type of site of disease relapse (log-rank test *p* = 0.9564).

**Table 1 cancers-11-00452-t001:** Clinical characteristics based on the pattern of first relapse.

Patients			Pattern of First Relapse			
		Total (105)	Regional (A) (58; 55.2%)	Regional + Distant (B) (15; 14.3%)	Visceral Only (C) (32; 30.5%)	*p* **
Median time to relapse (years. 25th–75th)		1.9 (0.7–3.8)	1.8 (0.7–3.8)	1.8 (0.5–5.7)	2.2 (0.8–3.8)	0.956
Gender	F	42 (40%)	27 (70.5%)	8 (53.3%)	7 (21.9%)	0.038 (C vs. A 0.06)
	M	63 (60%)	31 (29.5%)	7 (46.7%)	25 (78.1%)	
Age at diagnosis	Median (interval)	70 (22–88)	71 (28–87)	72 (34–86)	66 (22–88)	0.249
Primary site	Head/neck ^a^	16 (15.2%)	8 (13.8%)	5 (33.3%)	3 (9.4%)	0.030 (d vs. b 0.021)
	Trunk ^b^	41 (39.1%)	16 (27.6%)	5 (33.3%)	20 (62.5%)	
	Upper extremities ^c^	12 (11.4%)	8 (13.8%)	2 (13.4%)	2 (6.2%)	
	Lower extremities ^d^	36 (34.3%)	26 (44.8%)	3 (20.0%)	7 (21.9%)	
Histotype	Nodular	16 (15.2%)	8 (13.8%)	4 (26.7%)	4 (12.5%)	0.407
	Other	89 (84.8%)	50 (86.2%)	11 (73.3%)	28 (87.5%)	
Breslow	mm ± DS	3.7 ± 2.8	4.0 ± 2.8	4.8 ± 2.8	2.7 ± 1.9	0.005 (C vs. B *p* = 0.04)
Ulceration	Absent	59 (56.2%)	33 (56.9%)	4 (26.7%)	22 (68.7%)	0.025 (C vs. B *p* = 0.02)
	Present	46 (43.8%)	25 (43.1%)	11 (73.3%)	10 (38.3%)	
SLNB	Not performed	64 (60.9%)	39 (67.2%)	11 (73.3%)	14 (43.7%)	0.052
	Negative	41 (39.1%)	19 (32.8%)	4 (26.7%)	18 (56.2%)	
AJCC 2017	IB	24 (22.9%)	9 (15.5%)	2 (13.3%)	13 (40.6%)	<0.001 (C vs. A *p* = 0.03 C vs. B *p* = 0.12)
	IIA	31 (29.5%)	19 (32.8%)	5 (33.3%)	7 (21.9%)	
	IIB	29 (27.6%)	18 (31.0%)	0 (0.0%)	11 (34.4%)	
	IIC	21 (20.0%)	12 (20.7%)	8 (53.4%)	1 (3.1%)	
Mutational status (Sequenom)	WT*	32 (30.5%)	15 (25.9%)	6 (40.0%)	11 (34.4%)	0.470
	*BRAF* MUT	44 (41.9%)	25 (43.1%)	8 (53.3%)	11 (34.4%)	
	*NRAS* MUT	27 (25.7%)	17 (29.3%)	1 (6.7%)	9 (28.1%)	
	*KRAS* MUT	1 (0.9%)	1 (1.7%)	0 (0%)	0 (0%)	
	*PIK3CA* MUT	1 (0.9%)	0 (0%)	0 (0%)	1 (3.1%)	
Mutational status site	Primary	42 (40.0%)	14 (24.1%)	8 (53.4%)	20 (62.5%)	<0.001 (B vs. A, C vs. B <0.001)
	Regional mts	46 (43.8%)	39 (67.3%)	5 (33.3%)	2 (6.3%)	
	Distant mts	17 (16.2%)	5 (8.6%)	2 (13.3%)	10 (31.2%)	
*TERT* promoter mutations	WT	31 (29.5%)	20 (34.5%)	4 (26.7%)	7 (21.9%)	0.440
	−146 or −124 C > T	74 (70.5%)	38 (65.5%)	11 (73.3%)	25 (78.1%)	
*TERT* rs2853669	Absent	51 (48.6%)	28 (48.3%)	8 (53.3%)	15 (46.9%)	0.916
	Present	54 (51.4%)	30 (51.7%)	7 (46.7%)	17 (53.1%)	
*TERT/*Trunk site	WT/no trunk ^a^	24 (22.9%)	15 (25.9%)	3 (20.0%)	6 (18.8%)	0.0045 (D vs. B *p* = 0.01)
	−146 or −124 C > T mut/no trunk ^b^	40 (38.1%)	27 (46.5%)	7 (46.7%)	6 (18.8%)	
	WT/trunk ^c^	7 (6.7%)	5 (8.6%)	1 (6.7%)	1 (3.1%)	
	−146 or −124 C > T mut/trunk ^d^	34 (32.4%)	11 (18.9%)	4 (26.7%)	19 (59.4%)	

* One case D816H c-kit mutated; ** Bonferroni correction has been reported when significative. a: Head/neck; b: Trunk; c: Upper extremities; d: Lower extremities Bonferroni correction groups.

**Table 2 cancers-11-00452-t002:** *TERT* promoter mutations details based on pattern of first relapse.

Patients		Total (105)	Other Site of Progression (73; 55.4%)	Only Visceral (32; 44.6%)
*TERT* Promoter mutations	−124 C > T	37	24 (32.9%)	13 (40.6%)
	−146 C > T	37	25 (34.2%)	12 (37.5%)
	−57 A > C	2	2 (2.7%)	0 (0%)
	−125_124 CC > TT	2	2 (2.7%)	0 (0%)
	−139_138 CC > TT	2	1 (1.4%)	1 (1.4%)
	WT	25	19 (26.0%)	6 (18.8%)

**Table 3 cancers-11-00452-t003:** Clinical characteristics based on *TERT* promoter mutations.

Patients			*TERT* Promoter		*p* **
		Total	WT or Minor Mutations (31)	−146 or −124 C>T Mutation (74)	
Gender	F	42 (40%)	13 (41.9%)	29 (39.2%)	0.793
	M	63 (60%)	18 (58.1%)	45 (60.8%)	
Age at diagnosis	Median (interval)	70 (22–88)	72 (22–85)	68 (26–88)	0.950
Primary site	Head/neck ^a^	16 (15.2%)	5 (16.1%)	11 (14.9%)	0.010 (d vs. b 0.014 c vs. d 0.010)
	Trunk ^b^	41 (39.0%)	7 (22.6%)	34 (45.9%)	
	Upper extremities ^c^	12 (11.4%)	1 (3.2%)	11 (14.9%)	
	Lower extremities ^d^	36 (34.3%)	18 (58.1%)	18 (24.3%)	
Histotype	Nodular	16 (15.2%)	3 (9.7%)	13 (17.6%)	0.305
	Other	89 (84.8%)	28 (90.3%)	61 (82.4%)	
Breslow	mm ± DS	3.7 ± 2.8	4.1 ± 3.2	3.6 ± 2.6	0.362
Ulceration	Absent	59 (56.2%)	16 (51.6%)	43 (58.1%)	0.541
	Present	46 (43.8%)	15 (48.4%)	31 (41.9%)	
SLNB	Not performed	64 (60.9%)	19 (61.3%)	45 (60.8%)	0.963
	Negative	41 (39.1%)	12 (38.7%)	29 (39.2%)	
AJCC 2017	IB	24 (22.8%)	3 (9.7%)	21 (28.4%)	0.175
	IIA	31 (29.5%)	12 (38.7%)	19 (25.7%)	
	IIB	29 (27.6%)	10 (32.3%)	19 (25.7%)	
	IIC	21 (20.0%)	6 (19.3%)	15 (20.3%)	
Mutational status (Sequenom)	WT *	32 (30.5%)	12 (38.7%)	20 (27.0%)	0.187
	*BRAF* MUT	44 (41.9%)	8 (25.8%)	36 (48.7%)	
	*NRAS* MUT	27 (25.7%)	11 (35.5%)	16 (21.6%)	
	*KRAS* MUT	1 (0.95%)	0 (0%)	1 (1.35%)	
	*PIK3CA* MUT	1 (0.95%)	0 (0%)	1 (1.35%)	
Mutational status site	Primary	42 (40%)	13 (41.9%)	29 (39.2%)	0.496
	Regional mts	46 (43.8%)	15 (48.4%)	31 (41.9%)	
	Distant mts	17 (16.2%)	3 (9.7%)	14 (18.9%)	
*TERT* rs2853669	Absent	51 (48.6%)	16 (51.6%)	35 (47.3%)	0.687
	Present	54 (51.4%)	15 (48.4%)	39 (52.7%)	

* One case D816H *KIT* mutated, ** Bonferroni corrections were reported when significative. a: Head/neck; b: Trunk; c: Upper extremities; d: Lower extremities Bonferroni correction groups.

**Table 4 cancers-11-00452-t004:** Univariate logistic regression.

Variable		Only Visceral Site		
		OR	CI	*p*
Age (linear)		0.98	0.95–1.00	0.141
Gender (F vs. M)		3.28	1.26–8.55	0.015
Breslow (linear)		0.74	0.57–0.94	0.015
Ulceration (absent vs. present)		0.46	0.19–1.12	0.089
AJCC 2017 (IB vs. IIA, IIB, IIC)		0.26	0.09–0.67	0.006
NM versus other histotype		0.72	0.21–2.45	0.606
Trunk vs. other site		3.41	1.43–8.11	0.006
Mutational status (Mut vs. WT)		0.77	0.32–1.87	0.566
SLNB status (negative vs. not performed)		2.79	1.18–6.57	0.019
−146 or −124 C > T *TERT* promoter mutations		1.75	0.66–4.61	0.259
*TERT* rs2853669		1.10	0.48–2.53	0.818
*TERT* status and trunk association only in primary lesions	WT/no trunk	1		
	−146 or −124 C > T mut/no trunk	1.52	0.25–9.29	0.648
	WT/trunk	2.67	0.12–57.6	0.532
	−146 or −124 C > T mut/trunk	5.33	1.02–27.7	0.047
*TERT* status and trunk association in all	WT/no trunk	1		
	−146 or −124 C > T mut/no trunk	0.68	0.21–2.28	0.538
	WT/trunk	0.50	0.05–5.03	0.556
	−146 or −124 C > T mut/trunk	3.64	1.14–11.66	0.029
*TERT* mut + trunk site vs. other only in primary		4.00	1.09–14.62	0.036
*TERT* mut + trunk site vs. other		4.78	1.93–11.8	0.001

**Table 5 cancers-11-00452-t005:** Multivariate logistic regression.

Multivariable	Only Visceral Site		
	OR	CI	*p*
Age (linear)	0.99	0.96–1.03	0.684
Gender (F vs. M)	3.44	1.12–10.6	0.031
AJCC 2017 (IB vs. IIA, IIB, IIC)	0.22	0.07–0.67	0.008
SLNB status (negative vs. not performed)	3.05	1.06–8.78	0.039
*TERT* mut + trunk site vs. other	3.78	1.35–10.6	0.011

**Table 6 cancers-11-00452-t006:** Clinical characteristics of Istituto Valenciano de Oncologia dataset based on pattern of first relapse.

Patients		Total (83)	Other Site of Progression (46; 55.4%)	Only Visceral (37; 44.6%)	*p*
Gender	F	31 (37.3%)	21 (45.6%)	10 (27.0%)	0.081
	M	52 (62.7%)	25 (54.4%)	27 (73.0%)	
Age at diagnosis	Median (interval)	64 (21–87)	65 (26–87)	60 (21–84)	0.404
Primary site	Head/neck	22 (26.5%)	9 (19.6%)	13 (35.1%)	0.059
	Trunk	25 (30.1%)	5 (44.0%)	20 (37.8%)	
	Upper extremities	9 (10.8%)	2 (13.0%)	2 (8.1%)	
	Lower extremities	27 (32.5%)	3 (43.5%)	7 (18.9%)	
SLNB	Not performed	18 (21.7%)	10 (21.7%)	8 (21.6%)	0.990
	Negative	65 (78.3%)	4 (78.3%)	18 (78.4%)	
AJCC 2017	IB	24 (28.9%)	15 (32.6%)	9 (24.3%)	0.408
	IIA, IIB, IIC	59 (71.1%)	31 (67.4%)	28 (75.7%)	
*TERT/*Trunk site	WT/no trunk	34 (40.9%)	20 (43.5%)	14(37.8%)	0.149
	−146 or −124 C > T mut/no trunk	24 (28.9%)	15 (32.6%)	9 (24.3%)	
	WT/trunk	13 (15.7%)	8 (17.4%)	5 (13.5%)	
	−146 or −124 C > T mut/trunk	12 (14.5%)	3 (6.5%)	9 (24.3%)	

**Table 7 cancers-11-00452-t007:** Multivariate logistic regression in Istituto Valenciano de Oncologia dataset.

Multivariable Logistic Regression	Only Visceral Site		
	OR	CI	*p*
Age (linear)	0.99	0.97–1.02	0.856
Gender (F vs. M)	2.49	0.92–6.70	0.071
AJCC 2017 (IB vs. IIA, IIB, IIC)	0.99	0.32–3.07	0.997
SLNB status (negative vs. not performed)	1.26	0.41–3.86	0.682
*TERT* mut + trunk site vs. other	4.80	1.01–22.9	0.049

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
