# Peer review of "TERT Promoter Mutations are Associated with Visceral Spreading in Melanoma of the Trunk"

_cancers, 2019, doi:10.3390/cancers11040452_

Round 1
Reviewer 1 Report
The manuscript entitled “Usefulness of Routine Tert Promoter Mutation Analysis at Diagnosis for Trunk Melanomas” addresses the highly debated issue of the prognostic power of TERT promoter mutations. The study is well designed and performed, the results are presented and interpreted in a straightforward manner, the manuscript is well written. Statistical analyses are adequate. The authors analyze the association of TERT promoter mutations with metastasis in two independent cohorts where they find an association of these mutations with metastasis limited for primary lesions of the trunk.
The association described is significant and replicated in an independent dataset. Nonetheless this is probably not the last word on this issue since in the validation cohort, significance is borderline and the cohorts used do not allow for stratification according to the specific variant present. The authors should present a power calculation, based on their data, of the number of patients needed to analyze the variants separately.
The authors should change the title avoiding any statement on clinical application that does not appear justified by the present evidence.
The authors do not discuss why TERT promoter variants determine a higher risk of visceral metastases and why this is limited to primary lesions of the trunk. Do trunk primaries show a different mutational pattern than other lesions and could TERT promoter variants be associated with this pattern? The cohort size is likely too small to answer this question in a definite manner but the authors should discuss biological mechanisms potentially leading to the association observed in order to guide future studies that might deliver insight into the mechanisms behind this association.
The term “single nucleotide polymorphism” should be substituted by “single nucleotide variant” according to the guidelines for the interpretation of sequence variants of the American College of Genetics and Genomics.
Author Response
Response to Reviewer 1 Comments
We thank the reviewers for their careful reading of our manuscript and the many insightful comments and suggestions, which led us to an improvement of our work. Below we respond to the comments of each reviewer in detail, with our responses underlined. We also provide a revised manuscript that reflects their suggestions and comments
Reviewer 1
Comments and Suggestions for Authors
- The manuscript entitled “Usefulness of Routine Tert Promoter Mutation Analysis at Diagnosis for Trunk Melanomas” addresses the highly debated issue of the prognostic power of TERT promoter mutations. The study is well designed and performed, the results are presented and interpreted in a straightforward manner, the manuscript is well written. Statistical analyses are adequate. The authors analyze the association of TERT promoter mutations with metastasis in two independent cohorts where they find an association of these mutations with metastasis limited for primary lesions of the trunk.
We thank the reviewer for his/her appreciation to the subject of the study
- The association described is significant and replicated in an independent dataset. Nonetheless this is probably not the last word on this issue since in the validation cohort, significance is borderline and the cohorts used do not allow for stratification according to the specific variant present. The authors should present a power calculation, based on their data, of the number of patients needed to analyze the variants separately.
We thank the reviewer for this recommendation. We are conscious of the limitations of a real life study. For sure the sample size is limited as reported in the text. A sentence addressing the power calculation of the number of patients needed to analyze the variants separately has been added to the text. In particular, power calculation was performed using visceral specific recurrence as the primary end point. To reach 80% power and a two-sided α = 0.05 test, 230 melanomas would had have to be analyzed to detect a 5% significant difference in outcomes between -124 and -146 TERT mutations. Thus, since the power calculation did not permit a separate analyses, the two mutations have been combined together. Moreover, since in the majority of the papers these two major mutations are usually combined, we also decided to consider them together to make possible a proper comparison with previous results and a future metanalysis.
-The authors should change the title avoiding any statement on clinical application that does not appear justified by the present evidence.
The title has been changed in: Tert Promoter Mutations are associated to visceral spreading in melanoma of the trunk.
-The authors do not discuss why TERT promoter variants determine a higher risk of visceral metastases and why this is limited to primary lesions of the trunk. Do trunk primaries show a different mutational pattern than other lesions and could TERT promoter variants be associated with this pattern? The cohort size is likely too small to answer this question in a definite manner but the authors should discuss biological mechanisms potentially leading to the association observed in order to guide future studies that might deliver insight into the mechanisms behind this association.
In the discussion section, a sentence has been added about the biological mechanism related to TERT mutation and Trunk site and new references have been added on this specific point.
-The term “single nucleotide polymorphism” should be substituted by “single nucleotide variant” according to the guidelines for the interpretation of sequence variants of the American College of Genetics and Genomics.
The term has been substituted in the text as requested.

Reviewer 2 Report
Authors investigated mutants on cancer driver gene hTERT in melanoma patients and explored the relationship between mutations and melanoma metastasis. Authors found the mutants on hTERT promoter were associated with the trunk melanoma by visceral spreading. The study result is very interested and guidance for prediction melanoma metastasis and melanoma surgical. I suggest publishing this paper if the authors can answer the following questions.
1. In the manuscript, authors detected the -124 or -146 mutation on hTERT promoter. Although the other mutation on hTERT promoter, -57 A>C, only was found in 2 patients, it is still not appropriate to put these cases in the WT group in table 1 and 3. Based on the literature, -57 A>C mutant on hTERT also can increase hTERT expression level and be found in multiple cancer types, including melanoma. Alternatively, authors may list data of this mutant site with trunk melanoma. Is there data indicate whether this site associated with trunk melanoma? It will more helpful for understanding the regulation of these hTERT mutations in melanoma.
2. Authors detected the mutations on hTERT promoter by directly sequenced the PCR product. Actually, the PCR product is a mixture fragment from different alleles of a sample and contains the mutation and WT of hTERT promoter. It is possible only the WT allele, not the mutant allele is detected by sequencing. The ratio of WT vs mutation on hTERT promoter might be not correct. The better way to detect the mutation site on hTERT is subcloning the PCR product into the vector and sequencing the individual plasmid.
3. Authors need to check the manuscript again and amend the writing mistakes. Like in table 1, there are repeated numbers in the same cell.
Author Response
We thank the reviewers for their careful reading of our manuscript and the many insightful comments and suggestions, which led us to an improvement of our work. Below we respond to the comments of each reviewer in detail, with our responses underlined. We also provide a revised manuscript that reflects their suggestions and comments
Reviewer 2
Comments and Suggestions for Authors
Authors investigated mutants on cancer driver gene hTERT in melanoma patients and explored the relationship between mutations and melanoma metastasis. Authors found the mutants on hTERT promoter were associated with the trunk melanoma by visceral spreading. The study result is very interested and guidance for prediction melanoma metastasis and melanoma surgical. I suggest publishing this paper if the authors can answer the following questions.
We thank the reviewer for his/her appreciation on the topic of the study
1. In the manuscript, authors detected the -124 or -146 mutation on hTERT promoter. Although the other mutation on hTERT promoter, -57 A>C, only was found in 2 patients, it is still not appropriate to put these cases in the WT group in table 1 and 3. Based on the literature, -57 A>C mutant on hTERT also can increase hTERT expression level and be found in multiple cancer types, including melanoma. Alternatively, authors may list data of this mutant site with trunk melanoma. Is there data indicate whether this site associated with trunk melanoma? It will more helpful for understanding the regulation of these hTERT mutations in melanoma.
Data regarding -57 A>C mutations have been reported in the text. We are conscious and we have disclaimed the limitations of a real life study that determined a limited sample size. A sentence concerning the power calculation of the number of patients needed to analyze the variants separately has been reported in the text. In particular power calculation was performed using visceral specific recurrence as the primary end point. To reach 80% power and a two-sided α = 0.05 test, 230 melanomas would had have to be analyzed to detect a 5% significant difference in outcomes between -124 and -146 TERT mutation type. For this reason, the two mutations have been combined together in the analyses. Similarly, we agree with the reviewer that the -57 A>C variant of hTERT promoter increases hTERT expression level in multiple cancer types, but its frequence in our cohort did not permit further statistical considerations.
2. Authors detected the mutations on hTERT promoter by directly sequenced the PCR product. Actually, the PCR product is a mixture fragment from different alleles of a sample and contains the mutation and WT of hTERT promoter. It is possible only the WT allele, not the mutant allele is detected by sequencing. The ratio of WT vs mutation on hTERT promoter might be not correct. The better way to detect the mutation site on hTERT is subcloning the PCR product into the vector and sequencing the individual plasmid.
We agree with the reviewer’s comment. This limitation of using Sanger sequencing has been added in discussion and a new reference has been added on this specific point.
3. Authors need to check the manuscript again and amend the writing mistakes. Like in table 1, there are repeated numbers in the same cell.
We thank the reviewer for this recommendation. The manuscript and tables have been checked.

Reviewer 3 Report
The authors demonstrated molecular profiling of TERT promoter mutation in melanoma. This study is novel and interesting. Some major point should be revised and further experiment should be added. The specific comments are as following:
Abstract: detail result should be presented.
Introduction: The goal of this study should be emphasized. And, possible results or hypothesis should be presented.
Results: How about Kaplan-Meier survival analysis? Please provide figure.
Discussion: TERT promoter mutation showed some clinicopathological significance. Its implication or possible mechanism should be presented. It did not give us new information.
Check Errors (space, english).
Based on these concerns, I recommend the publication of this work with Minor revision.
Author Response
We thank the reviewers for their careful reading of our manuscript and the many insightful comments and suggestions, which led us to an improvement of our work. Below we respond to the comments of each reviewer in detail, with our responses underlined. We also provide a revised manuscript that reflects their suggestions and comments
Reviewer 3
Comments and Suggestions for Authors
The authors demonstrated molecular profiling of TERT promoter mutation in melanoma. This study is novel and interesting. Some major point should be revised and further experiment should be added. The specific comments are as following:
-Abstract: detail result should be presented.
We thank the reviewer for this recommendation. Detailed results have been added to the abstract section.
-Introduction: The goal of this study should be emphasized. And, possible results or hypothesis should be presented.
We thank the reviewer for this recommendation. The introduction has been modified accordingly.
-Results: How about Kaplan-Meier survival analysis? Please provide figure.
The Figure has been added to the submission.
Discussion: TERT promoter mutation showed some clinicopathological significance. Its implication or possible mechanism should be presented. It did not give us new information.
In discussion section a sentence has been added about the biological mechanism related to TERT mutation and Trunk site and new references has been added on this specific point.
- Check Errors (space, english).
Text and tables have been checked.
Based on these concerns, I recommend the publication of this work with Minor revision.
We thank the reviewer for his/her appreciation on the topic of the study

Round 2
Reviewer 2 Report
Authors addressed my questions in the revised manuscript. I suggest publishing this manuscript.